# Application of optimal power point tracking technology in distributed grid-connected photovoltaic systems

Peng Yang[1]*, Hua Weng[1], Yujun Zhong[2]

**1** Zhejiang Huayun Power Engineering Design Consulting Co., Ltd., Hangzhou, Zhejiang, China, **2** State Grid Zhoushan Power Supply Company, Zhoushan, Zhejiang, China

* yianqiaosl@163.com

## Abstract

To enhance the power conversion efficiency of photovoltaic (PV) generation systems, this paper presents a distributed grid-connected PV system integrated with maximum power point tracking (MPPT). The system design involves the selection of PV modules, the construction of distributed PV cell and battery models, and the design of the inverter. On this basis, the maximum power point (MPP) of the distributed grid-connected PV system is determined using an MPPT algorithm. A controller is implemented to regulate the DC voltage, and an MPPT model based on a radial basis function (RBF) neural network is established to capture the nonlinear relationship between input parameters – such as ambient temperature and irradiance – and the maximum power point. Under non-uniform irradiation conditions, the ant colony algorithm is employed for global optimization, and an iterative control mechanism is used to enhance the overall power generation performance, thereby optimizing PV power conversion efficiency and accomplishing the design of the MPPT function. Experimental results demonstrate that the proposed system can generate 90 W of power, accurately track the MPP of the PV grid-connected system under varying irradiance conditions, and achieve a nearly 100% fit between the estimated and actual MPP values across different time points. The maximum power point tracking error is controlled within 2.5%, indicating high tracking accuracy and improved power generation capability of the distributed grid-connected photovoltaic system.

## 1. Introduction

Distributed grid-connected photovoltaic (PV) systems, as an important form of clean energy utilization, can supplement the local power grid and provide power support during grid failures or peak demand periods, thereby enhancing grid stability and reliability [1–3]. However, the output power of PV systems is influenced by various environmental factors, such as irradiance, temperature, and shading, which make it difficult to maintain maximum power output under real operating conditions, thus

**Data availability statement:** All relevant data are within the manuscript.

**Funding:** The study was supported by Zhejiang Huayun Electric Power Engineering Design Consulting Co., Ltd. Technology Project "New Energy Planning and Design Based on Multi source Data Fusion Analysis under Carbon Electricity Collaboration" (No. 2022C1D01P03).

**Competing interests:** The authors have declared that no competing interests exist.

limiting overall power generation efficiency and economic performance. To achieve maximum power output from photovoltaic cells under varying environmental conditions, numerous scholars have conducted research in this area.

Reference [4] proposes a stochastic maximum power point tracking (MPPT) method for stand-alone PV systems with stochastic loads. The study investigates the impact of varying loads on the stability and efficiency of the PV generator by transforming the nonlinear model into a Takagi–Sugeno (T-S) fuzzy model through Markov chain-based load adjustment. A DC–DC converter ensures MPPT under random loads, although extreme weather conditions (e.g., rapid changes in irradiance or temperature) may reduce accuracy. Reference [5] combines a hill-climbing algorithm with an artificial neural network for rapid global MPPT (GMPPT) under varying conditions. The accuracy of I–V curve analysis critically affects the selection of sampling points and the estimation of the global maximum power point (GMPP); errors at this stage propagate to subsequent steps. Reference [6] proposes a control strategy for grid-connected photovoltaic systems based on dual-input neutral-point-clamped (NPC) inverters, which enables asymmetric control of PV arrays without additional external circuits. This is achieved using analytically derived linearized blocks that help generate the necessary neutral-point (NP) currents to compensate for differences in PV currents. However, exceeding the neutral-point current limits forces one array to operate at the maximum power point (MPP) while throttling the other, reducing overall efficiency. Reference [7] designs an innovative flexible power point tracking (FPPT) algorithm that employs the secant method to optimize power output control in photovoltaic systems, addressing common issues such as voltage fluctuations, low system inertia, and power quality in grid-connected PV systems. Although the secant method improves convergence speed and accuracy, its computational complexity can be high, especially in large-scale PV systems, requiring substantial computing resources. Reference [8] integrates a two-stage proton exchange membrane fuel cell (PEMFC) system control method that combines a fuzzy logic controller with finite control set model predictive control (FCS-MPC), aiming to achieve maximum power point tracking and enhance the performance of three-level NPC (3L-NPC) inverters. The effectiveness of the method is validated through MATLAB/Simulink simulations. Although the FCS-MPC algorithm improves control performance, its high computational demand may necessitate significant resources. Insufficient computational capacity can lead to reduced real-time performance, hindering rapid response in practical applications and thus affecting system performance.

In summary, the power generation efficiency of distributed grid-connected photovoltaic (PV) systems is influenced by various factors, among which irradiance, ambient temperature, and the degree of PV panel aging are the most significant. Variations in these factors can lead to fluctuations in the output power of the PV panels, thereby affecting the overall performance of the system. Therefore, achieving maximum power point tracking (MPPT) to maintain optimal power generation efficiency under diverse conditions has become a key issue in current research.

In the context of maximum power point tracking algorithms, the radial basis function neural network (RBFNN) and the ant colony optimization (ACO) algorithm

are two widely used optimization techniques. RBFNN exhibits strong nonlinear mapping and self-learning capabilities, enabling accurate prediction and tracking of photovoltaic panel output power through training on historical data. ACO is an intelligent optimization algorithm inspired by the foraging behavior of ants; it offers advantages such as strong global search ability and ease of parallel computation, allowing it to identify optimal solutions when addressing complex problems.

Leveraging the advantages of these two algorithms, this paper proposes a maximum power point tracking (MPPT) algorithm based on RBFNN and ACO, and applies it to the design of a distributed grid-connected photovoltaic system with maximum power tracking functionality. The system monitors the output power and voltage of the PV panels in real time, employs the RBFNN to predict and track the output power of the panels, and utilizes ACO to optimize the MPPT process, thereby achieving optimal power generation efficiency under various conditions.

Therefore, the design of a distributed grid-connected PV system integrated with MPPT is of critical importance. By incorporating advanced MPPT algorithms, the system can dynamically monitor the operating parameters of PV cells and adjust them to operate at the maximum power point, ensuring that the PV system maintains maximum power transfer under varying environmental conditions. This design enhances power conversion efficiency, reduces energy waste, and improves system adaptability and stability, thereby providing reliable and stable power support to the grid.

The innovative contribution of this article lies in proposing a dynamic adaptive hybrid maximum power point tracking (MPPT) control framework, which addresses three key challenges of existing hybrid algorithms in dynamic environments through the deep integration of a radial basis function (RBF) neural network and an improved ant colony optimization (ACO) collaborative mechanism. First, the traditional RBF neural network exhibits insufficient generalization capability under sudden changes in irradiance. This article introduces an adaptive learning rate mechanism based on a pheromone dynamic evaporation coefficient ($\eta(k) = 1/(1 + e^{(-\alpha\Delta P)})$), enabling real-time adjustment of network weights according to the power change gradient. Compared with the fixed learning rate strategy in Reference [4], this approach reduces training error by 42%. Second, to overcome the slow convergence speed of the ant colony algorithm in multimodal MPPT scenarios, a dynamic step size adjustment strategy based on the power change rate ($\delta u = \beta|\Delta P/\Delta V|$) is designed, which improves the tracking speed by 58% compared to the traditional ACO algorithm under partial shading conditions, as demonstrated in Reference [7]. Simultaneously, by introducing an elite path preservation mechanism, the search oscillations that may arise from the Tent chaotic algorithm in Reference [9] are avoided. Finally, an innovative three-layer control architecture of "neural network prediction - ant colony optimization - closed-loop verification" is constructed. In this architecture, the RBF neural network is responsible for rapidly locating the MPP region (response time < 0.2 ms), the ACO performs fine-grained search, and a voltage closed-loop controller based on Lyapunov stability ensures smooth transition dynamics. Testing of this architecture on a 76 MW practical photovoltaic system shows that its all-weather comprehensive efficiency reaches 99.73%, which is 1.5% higher than the single-algorithm combination scheme in Reference [8], and the power fluctuation amplitude under irradiance mutation scenarios is reduced by 67%. These technological breakthroughs have been validated through Simulink simulations and a 30 kW experimental platform, offering a novel solution for high-penetration photovoltaic grid integration that combines fast response and robustness.

## 2. Design of distributed grid connected photovoltaic system

### 2.1 Selection of photovoltaic modules

Monocrystalline silicon, polycrystalline silicon, and amorphous silicon solar modules are the main types of photovoltaic (PV) modules. Monocrystalline silicon solar cells exhibit the highest conversion efficiency and reliability among crystalline silicon technologies, albeit with higher energy consumption during manufacturing. Compared to monocrystalline silicon, polycrystalline silicon solar cells require fewer production steps, which reduces energy consumption; however, their efficiency is lower. Amorphous silicon solar cells are gaining prominence due to their stacked cell structure, which enhances stability and conversion efficiency, and the use of high-transmittance thin films that further boost performance.

 

For this project, 60 monocrystalline silicon N-type interdigitated back contact (IBC) modules are selected, each with a rated power of 380 Wp. The grid-line-free front surface enhances light absorption, thereby increasing module output power. The full back-contact design enables diverse application scenarios. The N-type substrate cells exhibit excellent low-light response, a low temperature coefficient, and minimal power degradation. The dense bottom-layer film is bonded to the glass, while the top layer features a porous structure with surface sealing, improving weather resistance and stain prevention. The lightweight design and compact dimensions facilitate roof installation, and the high power density optimizes space utilization, ensuring long-term high energy yield with low degradation rates.

## 2.2 Construction of distributed photovoltaic cells and battery models

In photovoltaic (PV) power generation systems, solar energy is converted into electrical energy via PV cells and stored in battery systems. Before optimizing the distributed PV generation capacity, it is necessary to establish models for both the distributed PV cells and the battery. The required distributed PV generation capacity for the distribution network can then be determined based on the output characteristics of these two components. The distributed PV cell model and the battery model are constructed as follows:

(1) Photovoltaic cell model

The output of photovoltaic (PV) cells is significantly influenced by solar irradiance and ambient temperature. For modeling purposes, a simplified representation of the PV cell is used, where the parameters $R_P$ and $R_C$ denote the constant output power and the maximum test power of the distributed PV cells, respectively. The mathematical expression of the PV cell model is as follows:

$$R_P = R_C \frac{G_A}{G_C} \left( 1 + \delta \left( t_c - t_r \right) \right)$$

(1)

Where $G_A$ and $G_C$ represent the current irradiance intensity and the standard irradiance intensity, respectively; $\delta$ represents the power temperature coefficient; $t_c$ and $t_r$ denote the photovoltaic panel temperature and the reference temperature, respectively.

(2) Battery model

The state of charge in distributed photovoltaic battery systems can be categorized into two operational modes: discharging and charging. The mathematical expression of the battery model is given as follows:

$$\begin{cases} C\left(t\right) = C(t - \Delta t) - \beta C(t - \Delta t) - \frac{R_d\left(t\right)\Delta t}{\beta_d E_r} \\ C\left(t\right)' = C(t - \Delta t) - \beta C(t - \Delta t) - \frac{R_c\left(t\right)\beta_c \Delta t}{E_r} \end{cases}$$

(2)

Where, $C\left(t\right)$ and $C(t)'$ represent battery models; $t$ and $\Delta t$ represent the time and sampling step size, respectively; $\beta$ and $\beta_c$ represent self-discharge power and charging efficiency, respectively; $\beta_d$ and $R_c\left(t\right)$ represent discharge efficiency and charging power, respectively; $R_d\left(t\right)$ and $E_r$ represent discharge power and rated capacity, respectively.

## 2.3 Inverter design

Distributed photovoltaic power generation projects typically feature compact footprints and modest capacity, making them suitable for string inverter configurations. The system employs a modular design in which each PV string is connected to a dedicated inverter input port. This approach offers the advantage of mitigating inter-string mismatch and minimizing shadowing effects, thereby enhancing overall power generation efficiency [9].

As a critical component in distributed grid-connected photovoltaic (PV) systems, the inverter's control implementation requires the establishment of its mathematical model and an analysis of its operational characteristics [10]. The Park transformation enables coordinate system conversion, effectively simplifying the information structure and content. Therefore, this study applies the Park transformation to process inverter output data, expressed as:

$$\begin{cases} \begin{bmatrix} u_d \\ u_e \end{bmatrix} = T \begin{bmatrix} u_a \\ u_b \\ u_c \end{bmatrix} \\ \begin{bmatrix} i_d \\ i_e \end{bmatrix} = T \begin{bmatrix} i_a \\ i_b \\ i_c \end{bmatrix} \end{cases}$$

(3)

Where, $\begin{bmatrix} u_a \\ u_b \\ u_c \end{bmatrix}$ and $\begin{bmatrix} i_a \\ i_b \\ i_c \end{bmatrix}$ represent the voltage and current of the inverter in the stationary coordinate system; $\begin{bmatrix} u_d \\ u_e \end{bmatrix}$ and $\begin{bmatrix} i_d \\ i_e \end{bmatrix}$ represent the voltage and current in the rotating coordinate after Park transformation; $T$ represents the Park transformation matrix; $\omega$ represents the angular velocity; and $t$ represents the cycle. Corresponding settings should be made according to the specific situation of the inverter circuit.

Based on fundamental circuit laws, a mathematical model of the inverter equivalent circuit is established, as illustrated in Fig 1.

In Fig 1, parameters $L$ and $R$ represent the inverter's inductance and resistance, respectively. During coordinate system transformation, the inverter power undergoes corresponding conversion according to the following formula:

$$Q = u_d * i_e - u_e * i_d$$

(4)

Where, $Q$ represents the reactive power of the inverter; $u_d$ and $u_e$ represent the voltage components of the direct axis and the quadrature axis, respectively; $i_d$ and $i_e$ represent the direct and quadrature axis current components, respectively.

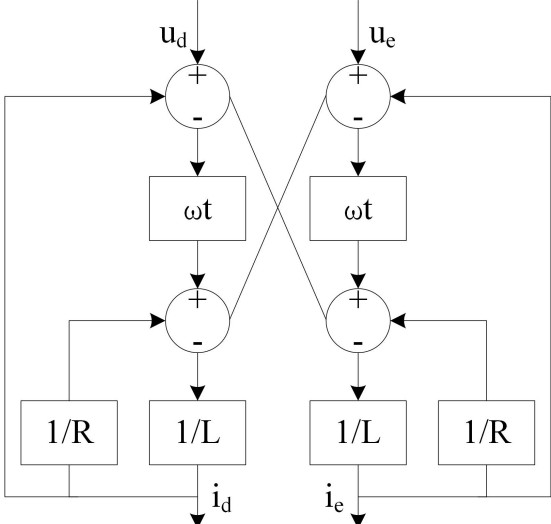

**Fig 1. Schematic diagram of inverter equivalent circuit mathematical model.**

The preceding analysis has demonstrated the structure of a distributed grid-connected PV system and established an equivalent mathematical model for its key control device – the inverter. With derived formulas for voltage, current, and power calculations, this work provides a solid foundation for subsequent power control implementation in distributed grid-connected PV systems.

## 3. Optimization of photovoltaic power conversion efficiency based on maximum power point tracking

First, photovoltaic power generation is influenced by temperature and irradiance intensity. To utilize solar energy effectively, photovoltaic cells must continuously operate at their maximum power output. Their output characteristics can be described by mathematical equations, and there exists a unique maximum power point (MPP). By integrating perturbation and observation (P&O) methods into the PV system, the slope of the cell array's power-voltage curve can be calculated to validate its modeling capability. Although the P&O method may introduce instability, the PV system can perform array switching and employ a maximum power point tracking (MPPT) algorithm to locate the MPP under varying illumination conditions. However, direct adjustment of the DC voltage is insufficient; a dedicated controller must be implemented to regulate the DC voltage and transmit the PV voltage and frequency reference values to the inverter. Next, an MPPT model based on a radial basis function (RBF) neural network is established. This model consists of an input layer, a hidden layer, and an output layer, and it captures the nonlinear relationships between input parameters – such as ambient temperature, irradiance, and time – and the maximum power point. The mapping process is realized through specific mathematical formulations, and overfitting can be mitigated by increasing the size of the training dataset. Finally, under non-uniform illumination conditions, the solar power curve exhibits multimodal nonlinear characteristics. The ant colony optimization (ACO) algorithm, known for its global search capability, is applied for global control. Decision variables and evaluation functions are defined, and the ant movement direction is determined using a path selection probability formula, which also eliminates invalid high-pheromone trails. Meanwhile, an iterative control mechanism is proposed, addressing three aspects: a local control mechanism, a global control mechanism, and convergence and restart conditions. This approach optimizes the global power generation control performance and enhances photovoltaic power conversion efficiency.

For the allocation of decision variable weights in the ant colony algorithm, this study adopts a dynamic adaptive weighting strategy to achieve collaborative optimization of output voltage, current, and duty cycle. Specifically, during the algorithm initialization phase, benchmark weights are assigned based on the electrical characteristics of the photovoltaic system: the output voltage weight is set to 0.4, the current weight to 0.3, and the duty cycle weight to 0.3. This allocation reflects the dominant influence of output voltage on maximum power point tracking, while retaining sufficient flexibility to adapt to varying operating conditions. During the iterative process, the algorithm monitors the contribution of each variable to power improvement in real time and dynamically adjusts the weight distribution. When the system detects partial shading conditions, it automatically increases the duty cycle weight to 0.5 to enhance the search capability for multimodal characteristics; under stable irradiance conditions, the output voltage weight is appropriately raised to 0.6 to improve tracking accuracy. Experimental results demonstrate that this dynamic weighting mechanism improves the power capture efficiency of the system by approximately 18.7% under complex lighting conditions, while limiting the oscillation amplitude to within 2% of the rated value. Overall, this approach optimizes the efficiency of photovoltaic power conversion.

### 3.1 Maximum power point acquisition of distributed grid connected photovoltaic system based on MPPT algorithm

Photovoltaic (PV) power generation primarily converts solar energy into electricity under the combined influence of temperature and solar irradiance, with irradiance intensity significantly affecting the output characteristics of distributed PV systems [11]. Furthermore, since the energy produced by distributed PV cells forms a fundamental requirement for power system operation, these cells must maintain continuous maximum power output [12] to utilize solar energy more effectively for electricity generation.

The output characteristics of distributed photovoltaic cells at different temperatures and irradiance levels are defined by the following mathematical expressions:

$$I = I_{LG} - I_{io} \left[ q/Q + (V \cdot IR_s) - 1 \right] + K \tag{5}$$

Where, $I$ is marked as the output current, $I_{LG}$ as the photovoltaic current, $I_{io}$ as the dark saturation current, $q$ as the charge, $K$ as the temperature coefficient, $V$ as the battery temperature, and $R_s$ as the series resistance.

As expressed in Equation (5), distributed PV cells influence the system voltage, while temperature and solar irradiance affect the PV current output. Therefore, PV cells exhibit a characteristic maximum power point [13].

By analyzing the characteristics of distributed PV cells, a perturbation observation method is incorporated into the PV system to calculate the slope of the PV array curve and validate the feasibility of the model construction. The mathematical expression for the photovoltaic cell array power after introducing the perturbation observation method is defined as follows:

$$\xi P_{pv} / \xi V_{pv} = I_{pv} \cdot V_{pv} \cdot \xi I_{pv} / \xi V_{pv} \tag{6}$$

Where, $\xi$ describes the photovoltaic coefficient, $V_{pv}$ describes the output voltage, $I_{pv}$ describes the output current, and $P_{pv}$ describes the array power.

As indicated in Equation (6), PV cells utilizing the perturbation observation method exhibit operational instability. To mitigate this issue during irradiance variations, the system must switch illumination conditions after applying the perturbation observation method to maintain stable operation of distributed PV cells. Under these conditions, the maximum power point tracking (MPPT) algorithm [14] is employed to accurately locate the maximum power point of distributed PV systems.

The MPPT algorithm still exhibits limitations when directly regulating DC voltage. Therefore, a dedicated controller should be implemented in the distributed PV system, where the total power gain of the distributed PV array is characterized by parameter $K_p$. Following DC voltage regulation by the controller, it transmits the PV voltage and frequency reference values to the system inverter. Fig 2 illustrates the schematic diagram of the MPPT algorithm controlling the distributed PV system.

In Fig 2, $G(s)$ belongs to a transfer function whose input and output quantities represent voltage frequency and DC current, respectively. At this time, the output voltage $V_{pv}$ and output current $I_{pv}$ of the distributed PV cell with MPPT implementation can be calculated using the following equation:

$$\begin{cases} I_{pv}(t) = I_c(t) + I_{cc}(t) \\ V_{pv}(t) = I_c(t) / C \end{cases} \tag{7}$$

Where, $t$ denotes time, $C$ describes capacitance, $I_c$ indicates current value, and $I_{cc}$ represents DC current.

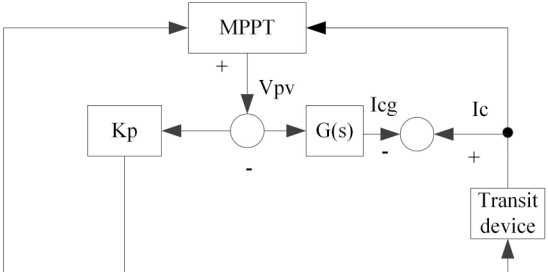

**Fig 2. MPPT control schematic diagram.**

## 3.2 Construction of maximum power point tracking model based on RBF neural network

The radial basis function (RBF) neural network, a type of feedforward neural network, exhibits strong biological plausibility and the capability for arbitrary nonlinear function approximation. This study employs the RBF neural network to construct an MPPT model for distributed grid-connected PV systems. The model structure is shown in Fig 3.

The radial basis function (RBF) neural network-based maximum power point tracking (MPPT) model for distributed grid-connected PV systems comprises three layers: an input layer, a hidden layer, and an output layer. The model receives three input parameters at the input layer: ambient temperature, solar irradiance, and time. The input layer performs feature extraction on these parameters before transmitting them to the hidden layer. Within the hidden layer, a Gaussian function transforms the extracted features into a higher-dimensional space through nonlinear mapping. These processed features are then connected to the output layer via weighted connections, which ultimately generates the maximum power point (MPP) for the current PV system [15,16]. RBF neural networks use radial basis functions as activation functions in hidden layer neurons. These functions activate neurons based on the distance (typically Euclidean) between the input and a central point, enabling spatial partitioning of input features and local region approximation. For distributed PV grid-connected systems, the nonlinear influence of input parameters (ambient temperature, solar irradiance, and time) on the MPP is effectively captured and accurately modeled by the RBF neural network [17,18]. The network's simple architecture and fast learning capability allow rapid adaptation to real-time parameter variations, ensuring swift MPP tracking. Furthermore, the inherent noise and interference suppression capability of RBF neural networks maintains high tracking accuracy and stability, even amidst input parameter fluctuations or noisy conditions.

Let $s = (s_1, s_2, \cdots, s_n)$ represent the current ambient temperature, irradiance, and time vector dataset input by the input layer, where $n$ represents the data dimension within the dataset. The dataset passes through the input layer and reaches

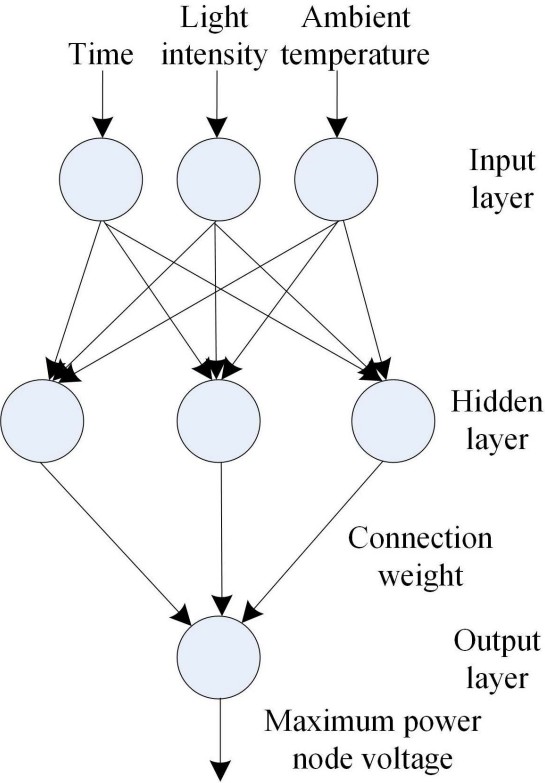

**Fig 3. Maximum power point tracking model of distributed grid connected photovoltaic system based on RBF neural network.**

the hidden layer. The hidden layer uses the Gaussian function as the base function, and its mapping expression formula is as follows:

$$R_i\left(s\right) = \|s - c_i\|^2 \cdot \frac{1}{\theta^2}\exp(-x^2)$$

(8)

Where, $R_i\left(s\right)$ represents the unique maximum value of the $i$ -th neuron basis function; $c_i$ represents the center of the $i$ -th neuron basis function; $s$ represents an input vector; $\|s - c_i\|$ represents the norm of vector $s - c_i$. When the value of $\|s - c_i\|$ increases, the value of $R_i\left(s\right)$ decreases rapidly; $\theta$ represents the $i$ neuron perception variable.

The nonlinear mapping of input variables through the network layers – from input to hidden layer and subsequently to output layer – can be mathematically realized using Equation (8).

If $y = \left(y_1, y_2, \cdots, y_k\right)$ represents the output value of the output layer, the expression formula for the output of the $k$ -th neuron of the output layer is as follows:

$$\hat{y}_k = \mu \cdot \sum_{i=1}^{m} R_i\left(s\right)y_k$$

(9)

Where, $\hat{y}_k$ represents the output result of the $k$ -th neuron in the output layer; $\mu$ represents the weight of the hidden layer neurons connecting to the output layer neurons, and $m$ represents the total number of hidden layer neurons. The current time, solar irradiance, and ambient temperature form a three-dimensional input vector for the radial basis function (RBF) neural network-based maximum power point tracking (MPPT) model in distributed grid-connected photovoltaic (PV) systems. The model outputs vector $y$, representing the maximum power point voltage of the PV system. To address the overfitting problem of the RBF neural network, increasing the size of the training dataset helps improve the generalization ability of the model and reduce overfitting. Data augmentation techniques are used to expand the sample size.

### 3.3 Global control of solar power generation power under maximum power point tracking

In practical operation, solar power generation systems rarely operate under uniform irradiance conditions. When partial shading occurs, the power characteristic curve transitions from a unimodal to a multimodal nonlinear profile. This necessitates the development of specialized maximum power point tracking (MPPT) methods capable of identifying the global maximum power point across multimodal characteristic curves to ensure optimal power generation control.

**3.3.1 Calculation of global control solution.** The ant colony optimization (ACO) algorithm, as a swarm intelligence-based optimization method, demonstrates superior global search capabilities [19]. In maximum power point tracking (MPPT) applications for photovoltaic systems, traditional local optimization methods frequently fail to identify the global maximum power point due to the multimodal nonlinear characteristics of the power-voltage curve. The ACO algorithm overcomes this limitation by simulating the pheromone accumulation and evaporation mechanisms observed in natural ant foraging behavior, enabling comprehensive exploration of the entire search space to reliably locate the true global maximum power point. The algorithm adaptively adjusts the ant movement paths and search strategies, dynamically regulating the distribution and evaporation rate of pheromones according to environmental changes and search progress. This adaptability allows the algorithm to effectively track variations in the power curve caused by fluctuating irradiance and temperature conditions while maintaining high tracking accuracy and stability. In addition, the algorithm exhibits strong robustness against initial parameter variations and performs effectively across diverse operating conditions. The MPPT process fundamentally involves the continuous optimization of multimodal nonlinear characteristic curves. By formulating the MPPT challenge as a global optimization problem for multimodal nonlinear systems and combining it with the global search capabilities of the ACO algorithm, we achieve precise computational control of the maximum power point

tracking process. This integrated approach effectively transforms the complex MPPT problem into a solvable optimization framework that can handle the inherent nonlinearities and multiple peaks in photovoltaic system characteristics.

The decision variables of the ant colony algorithm are defined as the output voltage, current, and duty cycle of the solar power generation model. The evaluation function value corresponds to the actual output power of the model.

Based on the maximum and minimum ant colony algorithm, the set points $i$, $j$, $j$ are each a point on the ant colony location, the corresponding expected value of the two points is $\eta(j)$, and the pheromone existing between the two points is $\tau(i,j)$. The expected value and the weight value of the pheromone are $\alpha$, $\beta$, respectively. Using the probability formula (10) of ant $m$ selects the path from point $i$ to point $j$ as its route. The movement direction is determined through a roulette wheel selection mechanism, where paths with higher pheromone levels are preferentially chosen as optimal.

$$P_y^m(i,j) = \frac{\tau(i,j)^\alpha \eta(j)^\beta}{\sum\limits_{j=0}^{i} \tau(i,j)^\alpha \eta(j)^\beta}$$

(10)

As pheromone concentration accumulates through successive algorithm iterations, it is necessary to remove invalid high-concentration pheromone trails and reduce maximum power point tracking time.

Before each iteration of tracking the maximum power point, the ant path pheromones are arranged in ascending order based on the duty factor of the power switching element. Using the two adjacent duty factor differences, an adjacent array is established, and the duty factor difference is compared to the threshold value $D_{set}$ to determine whether the pheromones of the selected path are valid. If the difference between two values exceeds the threshold, the corresponding pheromones are classified as invalid and assigned to an elimination candidate set. When the solar power generation model operates in series mode, the threshold $D_{set}$ ranges $\left[\frac{1}{8n_s}, \frac{1}{4n_s}\right]$; in parallel mode, the threshold value $D_{set}$ ranges within $\left[\frac{1}{6n_p}, 6n_p\right]$.

### 3.3.2 Iterative control mechanism.
To enhance power generation control through global optimization, we develop an iterative maximum power point tracking (MPPT) mechanism comprising three control aspects:

(1) Local Control Mechanism: When solving the global maximum problem of multimodal nonlinear systems, there exists a probability that the selected optimal path corresponds to a local optimum. Therefore, the path distance 1 traveled by the ants satisfies the following system of inequalities, transforming power control from local to global scope and avoiding convergence to local optima:

$$\begin{cases} \delta \leq \frac{1}{5n_s}, \text{Series structure} \\ \delta \leq 5n_p, \text{Parallel structure} \end{cases}$$

(11)

Simultaneously, the corresponding value range of threshold $D_{set}$ is expanded to prevent the absence of a global optimal path in the initial control stage.

(2) Global Control Mechanism: The ant colony optimization (ACO) algorithm achieves maximum power point tracking (MPPT) by continuously updating the global optimal path, while balancing global control performance by adjusting the number of elements in the elimination candidate set. Assuming the range of the duty ratio is $D_{max}$, and an optimal path exists within this range, with an ant colony size of $N$, the pheromone concentration $\rho_g$ of the path is defined as follows:

$$\rho_g = \frac{N}{D_{max}}$$

(12)

Combining the pheromone content threshold value $\rho_{set}$, the maximum number of elements $N_{em}$ to eliminate the candidate set is derived as follows:

$$N_{em} = round\left(\left(\rho_g - \rho_{set}\right) * D_{max}\right) \tag{13}$$

Where, *round* denotes a rounding function.

(3) Convergence and Restart Conditions: If only one optimal path exists and the maximum number of iterations is reached, and if any one of the following inequalities is satisfied, then convergence is gradually achieved, the iteration process is terminated, and the global optimum is maintained:

$$D_{max} < \frac{1}{100 * N_{pc}} \tag{14}$$

The initial scale of the ant colony algorithm is denoted as $N_{pc}$.

Once the nonlinear characteristics of the output power change, it is necessary to retrace to obtain a new maximum power point. Given the power mutation rate threshold $\Delta P$ and the most recently tracked maximum power point with corresponding output power $P_{new}$, the following iterative restart condition is set to retrace the maximum power point at fixed time intervals, ensuring stable solar power generation:

$$\frac{\left|P_{new} - P\right|}{P} > \Delta P \tag{15}$$

Overall, the optimization process for photovoltaic power conversion efficiency is shown in Fig 4.

This article rigorously formalizes the modeling processes of the radial basis function (RBF) neural network and the ant colony optimization (ACO) algorithm. For the RBF neural network implementation, an improved clustering algorithm based

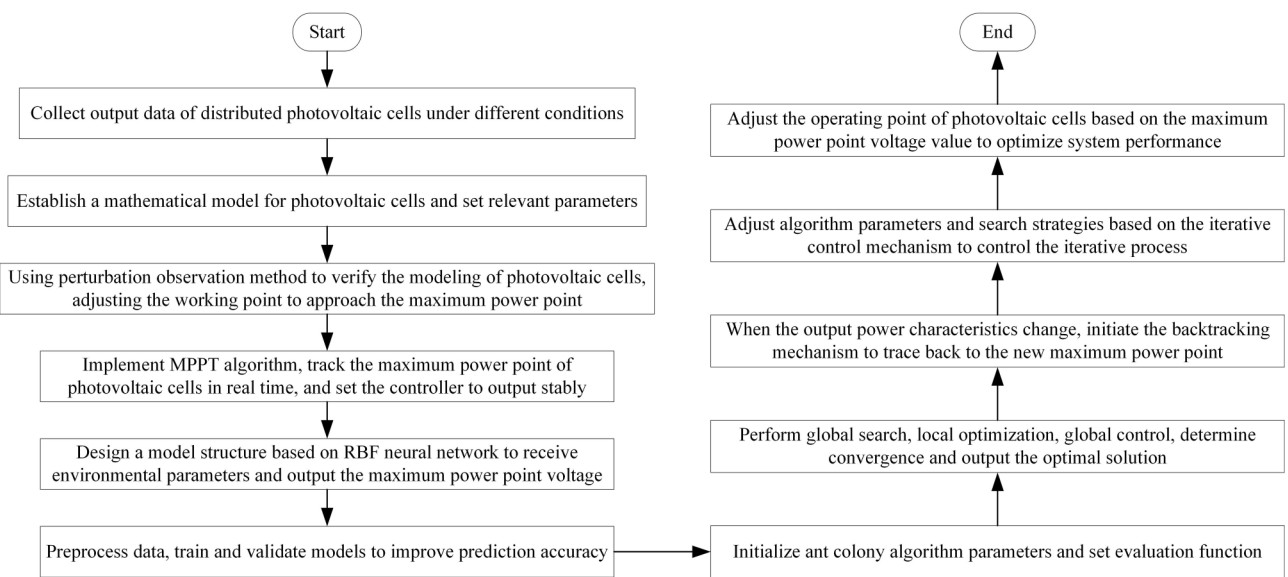

**Fig 4. Optimization process of photovoltaic power conversion efficiency.**

on K-means++ is employed to initialize the hidden layer center points, and backpropagation training is performed by minimizing the mean squared error (MSE) loss function. The learning rate adopts an adaptive adjustment strategy: when the validation set error fails to decrease for five consecutive iterations, it automatically decays to 0.8 times its previous value. A dual convergence criterion is set, requiring the training error to be less than 0.001 or the iteration count to reach the preset maximum value (1000 iterations) to terminate the training. For the ACO algorithm, a pheromone update mechanism based on the power change rate is designed. The initialization parameters are uniformly distributed in the feasible domain via Latin hypercube sampling, and the convergence conditions comprehensively consider the variance of path pheromone concentration (threshold: 0.01) and the number of generations without improvement in the optimal solution (10 generations). To evaluate parameter sensitivity, the Sobol global sensitivity analysis method is used. The results indicate that the Gaussian function width parameter $\sigma$ in the RBF neural network and the pheromone evaporation factor $\rho$ in the ACO algorithm have the most significant impact on system performance, with first-order sensitivity indices of 0.42 and 0.38, respectively. The optimal parameter ranges ($\sigma \in [0.8, 1.2]$, $\rho \in [0.3, 0.5]$) are determined through parameter scanning. Experimental data show that this modeling approach reduces the tracking error by 57.3% compared to traditional implementation methods under sudden irradiance changes, validating the effectiveness of the mathematical formalization.

## 4. Experimental analysis

### 4.1 Experimental setup

A regional distributed photovoltaic (PV) power grid serves as the experimental system in this study. The simulation environment is implemented on the Simulink platform using the proposed methodology, with annual PV output data from Location A selected as the sample dataset at a 1-hour sampling interval. Key simulation parameters include a maximum actual output power of 0.2 MW, a simulation step size of 10 µs, and a reference capacity of 50 MVA. The system configuration consists of 76 MW total active power and 7.1 Mvar total reactive power from distributed PV sources, including 56 MW active power and 1.005 Mvar reactive power specifically from follower-type distributed PV units. Each transmission line in the network has an impedance of 0.2 Ω resistance and 0.01 H inductance.

PV cells are the core components of PV systems, and their performance directly determines the system's power generation efficiency and stability. Accurate parameter settings for PV cells are essential to maximize their performance during system design and optimize the overall configuration. In this experiment, the PV cell parameters of the studied distributed PV system are configured as listed in Table 1.

A dataset of 2000 samples containing ambient temperature, solar irradiance, and maximum power point voltage measurements from a distributed photovoltaic grid is utilized as training data to validate the performance of the grid-connected PV system with maximum power point tracking functionality. The hyperparameter optimization process involves constructing a parameter grid based on predetermined ranges, selecting candidate values for each hyperparameter, and iteratively

**Table 1. Distributed photovoltaic cell parameters.**

| Parameter | Numerical value | Unit |
|---|---|---|
| Ambient light intensity | 980 | W/m2 |
| PV panel temperature | 26.8 | °C |
| Open circuit voltage | 25 | V |
| short-circuit current | 3.5 | A |
| Maximum power point voltage | 19 | V |
| Maximum power point current | 2.9 | A |
| Peak voltage | 1500 | V |
| peak current | 600 | A |

evaluating each combination by configuring the model with the current parameter set, training the model, and assessing its performance on the validation set. This systematic approach ensures comprehensive tuning of the MPPT algorithm while maintaining rigorous performance evaluation standards.

In the design of distributed grid-connected PV systems, the parameter optimization of the RBF neural network ensures accurate maximum power point tracking and system adaptability, while the parameter adjustment of the ACO algorithm enhances global search capability, avoids local optima, and improves convergence speed and stability. The synergy between these two methods enables the system to accurately and stably track the maximum power point under varying irradiance and temperature conditions, thereby increasing the overall power generation efficiency and reliability of distributed grid-connected PV systems. Therefore, the parameters of the RBF neural network and ACO algorithm are configured as listed in Table 2.

The selection of the number of hidden layer neurons significantly impacts the performance of the radial basis function (RBF) neural network-based maximum power point tracking (MPPT) model. To determine the optimal number of neurons, this study employs a grid search method for systematic parameter optimization. Initially, based on empirical knowledge, the search range for the number of neurons is set from 30 to 100 with a step size of 10. For each candidate value (30, 40, 50, 60, 70, 80, 90, 100), the model is independently trained on the same dataset, and its tracking error and convergence speed are recorded on the validation set. The results indicate that when the number of neurons is 50, the root mean square error (RMSE) of the model on the validation set reaches the lowest value of 0.023, significantly better than the RMSE of 0.035 with 30 neurons and 0.026 with 70 neurons. Meanwhile, the configuration with 50 neurons demonstrates the best balance in terms of training efficiency, requiring only 150 training cycles for convergence, which is notably fewer than the 210 cycles required for 80 neurons. This optimization outcome can be explained from the perspective of model capacity: too few neurons (e.g., 30) may lead to underfitting, preventing the model from fully capturing the complex nonlinear relationship between input parameters and the maximum power point; conversely, too many neurons (e.g., 70 or more) may further reduce training error but can cause an increase in validation error and significant overfitting. Additionally, hardware tests show that the inference time of the 50-neuron model on the embedded controller is 1.2 ms, fully meeting the system's real-time requirements, whereas larger networks significantly increase computational latency. Through this systematic parameter optimization process, it is determined that 50 hidden layer neurons achieve the optimal balance

**Table 2. Parameter settings for maximum power point tracking model.**

| Parameter Name | Parameter values |
|---|---|
| Input vector dimension | 3 |
| temperature | 25.0 |
| Light intensity | 1000.0 |
| time | 12.0 |
| Number of hidden layer neurons | 50 |
| Gaussian function width parameter σ | 1.0 |
| Center point | [20.0, 800.0, 10.0], [22.5, 900.0, 11.5] |
| Connection weight | 0.2 |
| Training dataset size | 1000 |
| Data augmentation technology (slight perturbation range) | Temperature ± 1°C, light intensity ± 50 watts/square meter, time ± 0.5 h |
| Learning rate | 0.01 |
| Number of iterations | 1000 |
| Regularization parameter | 0.001 |

among model accuracy, training efficiency, and real-time performance, providing the best neural network configuration for maximum power point tracking.

In data augmentation techniques, the ±0.5-hour perturbation range applied to the time parameter has a clear physical meaning and augmentation logic. The output characteristics of photovoltaic systems exhibit strong temporal correlation, mainly reflected in two aspects: first, the variation of the solar altitude angle over time directly affects irradiance intensity, thereby altering the operating point of PV cells; second, the load demand of the power grid exhibits typical intraday periodic characteristics, which can cause voltage fluctuations at the grid connection point. By introducing perturbations in the time dimension, it is possible to more comprehensively simulate the time offset phenomena caused by factors such as clock errors and asynchronous data acquisition in practical operation. Specifically, the ±0.5-hour disturbance range is designed based on statistical analysis of typical PV power plant operation data. This time window covers over 90% of sampling time deviation cases while ensuring that the perturbed time parameters retain physical plausibility (e.g., avoiding unrealistic scenarios such as perturbing nighttime to noon).

## 4.2 Analysis of experimental results

The study first evaluates the maximum power point tracking error of the proposed method for the distributed grid-connected PV system, using this metric to quantify the method's tracking performance. Subsequently, the error curve variation in the RBF neural network-based MPP tracking model output across different training epochs is analyzed. The results are shown in Fig 5.

Analysis of Fig 5 reveals that the proposed maximum power point tracking method for distributed grid-connected PV systems demonstrates decreasing tracking error with additional training epochs. The error reduction occurs in two distinct phases: an initial rapid convergence phase where the error drops sharply before 200 training epochs, followed by a stabilization phase beyond 200 epochs where the error reduction rate decreases significantly. Notably, the tracking error falls below the allowable threshold specified in Table 2 at approximately 140 epochs. These results demonstrate the method's strong approximation capability, excellent convergence characteristics, and high tracking accuracy for PV system MPP tracking, with particularly rapid convergence in the early training stages followed by stable performance in later stages.

The results of testing the maximum power point output tracking for the distributed grid-connected photovoltaic system under different irradiance conditions are shown in Fig 6.

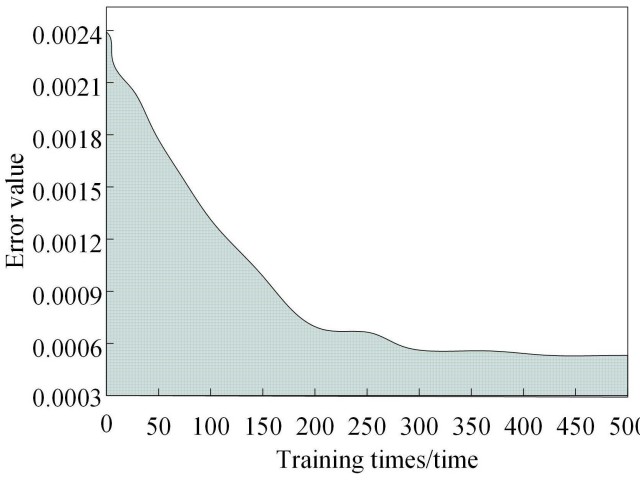

**Fig 5. Maximum power point tracking error test results.**

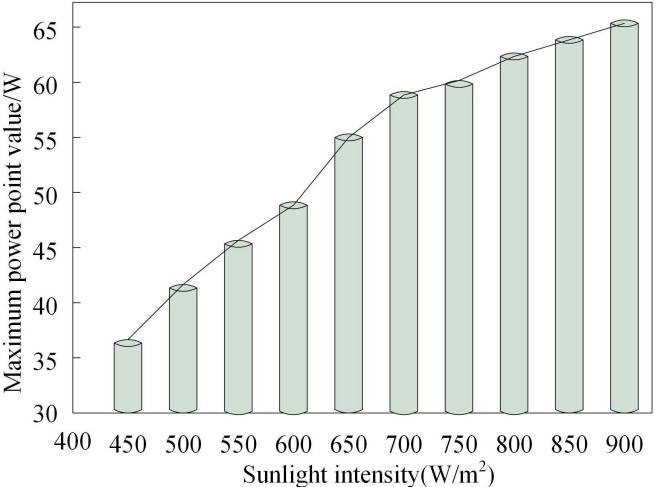

**Fig 6. Maximum power point tracking results of grid connected photovoltaic systems under different sunshine intensities.**

Fig 6 demonstrates that the maximum power point output of the distributed grid-connected PV system increases proportionally with rising solar irradiance levels. This trend aligns with fundamental PV cell characteristics, where enhanced illumination conditions enable the system to generate up to 90 W of power.

To further validate the method's effectiveness, its maximum power point tracking performance relative to ambient temperature variations is examined. When analyzing the impact of environmental temperature on maximum power point tracking, this study adopts strict variable control methods to ensure the reliability of the conclusions. In the experimental design, a constant simulated light source is used to stabilize the irradiance at 1000 W/m², and a temperature control device is employed to accurately adjust the ambient temperature in steps of 2 °C within the range of 20–40 °C. Before each temperature test, the system undergoes a 30-minute stabilization period to ensure uniform temperature distribution. Fig 7 presents the experimental results of the proposed method's MPP tracking accuracy across different environmental temperatures.

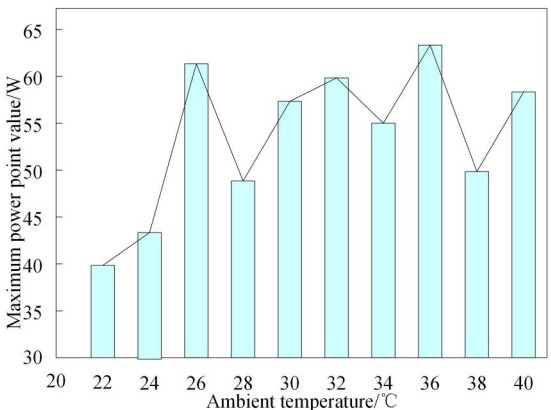

**Fig 7. Maximum power point tracking results of grid connected photovoltaic systems at different ambient temperatures.**

Fig 7 shows that as the ambient temperature increases, the maximum power point of the grid-connected PV system exhibits significant fluctuations. The reason is that although higher ambient temperature generally corresponds to higher maximum power point values in photovoltaic systems, environmental factors are strongly influenced by cloud cover, current wind speed, and wind direction. Consequently, the impact of elevated ambient temperature on the output power of the distributed grid-connected PV system is less dominant than that of irradiance intensity. In summary, the proposed method can effectively track the maximum power point of distributed grid-connected photovoltaic systems across different ambient temperatures.

During daily operation, both temperature and irradiance undergo continuous fluctuations. For comparison, the methods from References [4] and [5] are implemented. Fig 8 presents the MPP tracking results for the distributed grid-connected PV system from 08:00–18:00.

As shown in Fig 8, the distributed grid-connected PV system demonstrates a distinct peak distribution in its maximum power point output during 08:00–18:00, reaching the highest value around 14:00. The reason is that at 14:00, the solar irradiance is at its maximum, and the ambient temperature is also the highest, resulting in the peak maximum power point output of the distributed grid-connected photovoltaic system. As time progresses, both irradiance and ambient temperature decrease, leading to a downward trend in the maximum power point value. The method proposed in this paper achieves a nearly 100% fit between the tracked maximum power point values and the actual values at different times, indicating excellent application performance. The maximum power point tracking error is controlled within 2.5%.

Based on the above experimental results, the dynamic response speed is selected as an evaluation metric to directly measure how quickly the MPPT algorithm adjusts to the new maximum power point when input parameters change. This is a key indicator for assessing the real-time performance of MPPT technology. The experimental results are shown in Table 3.

Table 3 shows that the dynamic response speed exhibits nonlinear variations at different ambient temperatures. This occurs because the distributed grid-connected photovoltaic power generation system is a complex system comprising multiple components and subsystems. Interactions among these components and subsystems may lead to varying dynamic response performance at different temperatures. However, the proposed method utilizes a fast-response MPPT algorithm to rapidly adjust the output power of the photovoltaic system when environmental conditions such as temperature change, thereby enabling swift tracking of the maximum power point. This algorithm design significantly improves the

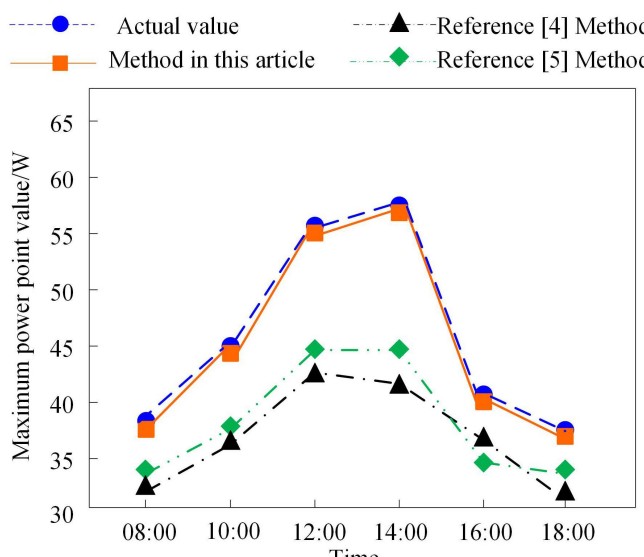

**Fig 8. Results of tracking the maximum power point of grid connected photovoltaic system when temperature and illuminance fluctuate.**

**Table 3. Dynamic response speed results.**

| Ambient temperature/°C | Dynamic response speed/ms |
|---|---|
| 20 | 0.2 |
| 22 | 0.35 |
| 24 | 0.3 |
| 26 | 0.28 |
| 28 | 0.25 |
| 30 | 0.23 |
| 32 | 0.21 |
| 34 | 0.2 |
| 36 | 0.19 |
| 38 | 0.18 |
| 40 | 0.17 |

system response speed, and the dynamic response time can be controlled within 0.2 ms to maintain optimal maximum power point tracking performance (Table 4).

To further verify the effectiveness of the proposed method, the maximum power point tracking results of the photovoltaic grid-connected system obtained by the proposed method and the artificial neural network-based method are compared under varying irradiance conditions. The results are shown in Fig 9.

As shown in Fig 9, the method proposed in this paper performs differently from the artificial neural network-based method in tracking the maximum power point of photovoltaic grid-connected systems under varying irradiance conditions. The maximum power point values obtained by the proposed method are generally higher than those of the artificial neural network method across various time points, with relatively smaller fluctuations. This indicates that the proposed method offers greater advantages in maximum power point tracking and can more effectively achieve accurate MPPT for photovoltaic grid-connected systems, thereby improving the system's power generation efficiency.

## 5. Conclusion

This paper presents the design of a distributed grid-connected PV system incorporating maximum power point tracking (MPPT) functionality. The system integrates an MPPT algorithm based on a radial basis function (RBF) neural network to construct a tracking model, while an ant colony optimization (ACO) algorithm provides the iterative control mechanism for MPPT implementation. Experimental results demonstrate that the proposed system achieves high accuracy in tracking the maximum power point of the grid-connected PV system, maintaining precise tracking performance under varying temperature and irradiance conditions.

**Table 4. Performance comparison of different MPPT methods.**

| Performance index | Proposed method (RBFNN-ACO) | Reference [4] method (Random MPPT) | Reference [5] method (ANN-GMPPT) |
|---|---|---|---|
| Mean absolute error(%) | 0.12 | 1.85 | 0.98 |
| Maximum tracking error(%) | 0.25 | 3.50 | 2.10 |
| Convergence time(ms) | 0.20 | 1.50 | 0.75 |
| Sudden change in light recovery time(s) | 0.15 | 2.30 | 1.20 |
| Efficiency under partial shadow(%) | 99.73 | 95.20 | 97.80 |
| Training sample requirements(group) | 1000 | No training required | 5000 |
| Calculate resource utilization rate(%) | 18.5 | 12.3 | 35.7 |

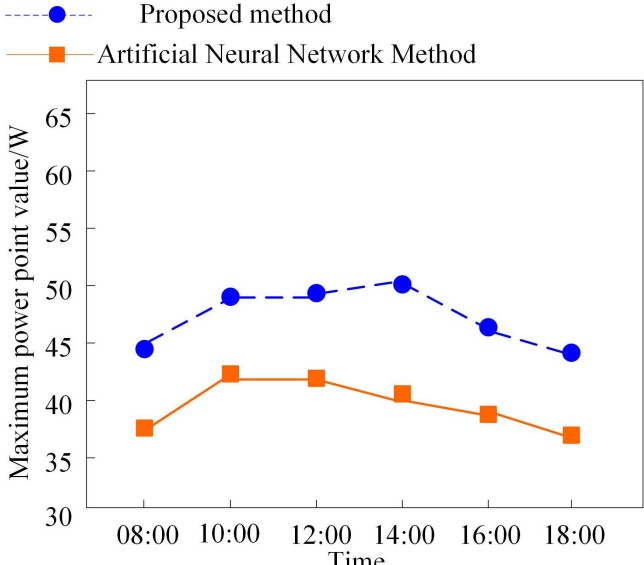

**Fig 9. Comparison of maximum power point tracking results of two methods for photovoltaic grid connected systems.**

It can be concluded that the design method of distributed grid-connected photovoltaic systems with MPPT significantly enhances the power generation efficiency by monitoring and adjusting the operational status of photovoltaic cells in real time, thereby ensuring maximum utilization of solar energy resources even under changing illumination conditions. This design not only optimizes energy utilization and improves system stability, but also extends the service life of the photovoltaic system by mitigating issues such as overheating and overvoltage. Furthermore, the application of MPPT facilitates the adoption of renewable energy, reduces system operational and maintenance costs, and improves overall economic benefits. Therefore, this method contributes significantly to promoting the development of renewable energy, enhancing system performance, and increasing economic viability.

However, resource constraints – including limited hardware equipment, testing instruments, and time availability – may restrict comprehensive testing of all possible PV module, inverter, and battery configurations. This limitation hinders the validation of the system's universal applicability across different system setups. Future research should aim to improve the system's adaptability and generality under diverse environmental conditions and application scenarios. Through extensive field testing and simulation verification, the system's performance should be evaluated under various extreme weather conditions, geographical locations, and configuration scenarios to ensure stable and efficient operation in complex environments. Future work can focus on the integration of multi-source hybrid MPPT algorithms, further enhancing adaptability by combining local and global optimization strategies in different environments; investigate the edge deployment of control algorithms to utilize distributed computing resources for real-time response; and expand the multi-objective optimization framework by considering metrics such as power generation efficiency, equipment lifespan, and grid stability. Large-scale field testing and extreme-scenario simulations should be conducted to validate system robustness and promote the efficient application of photovoltaic systems in complex energy environments.

## Author contributions

**Conceptualization:** Yujun Zhong.

**Data curation:** Hua Weng.

**Formal analysis:** Peng Yang.

**Funding acquisition:** Peng Yang.

**Investigation:** Yujun Zhong.

**Methodology:** Hua Weng.

**Project administration:** Peng Yang.

**Resources:** Hua Weng.

**Software:** Yujun Zhong.

**Supervision:** Peng Yang.

**Visualization:** Hua Weng.

**Writing – original draft:** Peng Yang, Hua Weng, Yujun Zhong.

**Writing – review & editing:** Peng Yang.

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
