## [Decision Letter · Decision Letter 0]

30 Jul 2025

Dear Dr. Yang,

Thank you for submitting your manuscript to PLOS ONE. After careful consideration, we feel that it has merit but does not fully meet PLOS ONE’s publication criteria as it currently stands. Therefore, we invite you to submit a revised version of the manuscript that addresses the points raised during the review process.

We look forward to receiving your revised manuscript.

Kind regards,

Zhengmao Li

Academic Editor

PLOS ONE

Journal Requirements:

“The study was supported by Zhejiang Huayun Electric Power Engineering Design Consulting Co., Ltd. Technology Project “New Energy Planning and Design Based on Multi source Data Fusion Analysis under Carbon Electricity Collaboration” (No. 2022C1D01P03).”

Reviewers' comments:

Reviewer's Responses to Questions

**Comments to the Author**

1. Is the manuscript technically sound, and do the data support the conclusions?

Reviewer #1: Yes

Reviewer #2: Partly

2. Has the statistical analysis been performed appropriately and rigorously?

Reviewer #1: Yes

Reviewer #2: I Don't Know

3. Have the authors made all data underlying the findings in their manuscript fully available?

Reviewer #1: Yes

Reviewer #2: Yes

4. Is the manuscript presented in an intelligible fashion and written in standard English?

Reviewer #1: Yes

Reviewer #2: Yes

Reviewer #1: This manuscript presents a hybrid maximum power point tracking (MPPT) approach that integrates radial basis function neural networks (RBFNN) and ant colony optimization (ACO) for distributed grid-connected photovoltaic systems. However, the manuscript requires substantial improvements in terms of methodological clarity, technical rigor, validation depth, and presentation quality before it can be considered for publication. The following comments are provided to assist the authors in strengthening their work:

1. The references are relevant but not cohesively organized. The authors are advised to categorize and compare existing MPPT techniques (e.g., P&O, ANN, PSO, FLC, ACO) to better position their contribution within the research landscape.

2. While combining RBF neural networks and ant colony optimization (ACO) for MPPT is practical, the concept is not new. The manuscript should clarify its novel contributions over prior works that employ similar hybrid algorithms.

3. The modeling of the RBF and ACO algorithms lacks rigor. The training procedures, loss functions, convergence criteria, and parameter initialization should be mathematically formalized and supported with sensitivity analysis.

4. Although visual comparisons are made with Ref [4] and [5], there is no tabulated performance comparison in terms of tracking efficiency, average absolute error, or convergence time. Including a summary table would better demonstrate the performance gain.

5. The manuscript contains numerous grammatical errors, awkward phrasings, and repetitive wording. Professional English editing is strongly recommended to meet journal language quality standards.

6. The conclusion mentions general limitations but fails to outline specific next steps. Suggestions such as integrating multi-source hybrid MPPT, edge deployment of control algorithms, or expanding to multi-objective optimization would strengthen the outlook.

Reviewer #2: The research direction of this paper has certain value, but there are significant flaws from theoretical derivation to experimental verification, with vague innovation points, failing to meet the publication standards.

1. The statement in the abstract that "fitting degree between the maximum power point values at different times and the actual values is close to 100%" is absolute, without providing a specific error range, violating scientific rigor.

2. The keywords section was initially empty, and the later supplemented ones such as "maximum power tracking; distributed grid-connected photovoltaic system" do not cover core algorithms (such as RBF neural network, ant colony algorithm), making the keyword setting unreasonable.

3. In 1-40, \(R_P\) is defined as output power and \(R_C\) as maximum test power, both with power units. In the formula \(R_{P}=\frac{R_{C} G_{A}}{G_{C}}\left(1+\delta\left(t_{c}-t_{r}\right)\right)\), \(G_A\) and \(G_C\) are light intensity (unit: W/m²). Direct multiplication of power and light intensity leads to mismatched physical dimensions, resulting in formula derivation errors.

4. In the inverter model part, the formulas from 1-48 to 1-53 are fragmented, with excessive "... ...", and key derivation steps are missing, making it impossible to reproduce the model construction process.

5. The number of neurons in the hidden layer of the RBF neural network is set to 50, without comparison with other numbers (such as 30, 70), and without explaining why 50 is the optimal choice, lacking the parameter optimization process.

6. In 1-125, for the data augmentation technology "time ± 0.5 h", the impact of time perturbation as an input parameter on the maximum power point is not analyzed, and the augmentation logic is unclear.

7. The decision variables of the ant colony algorithm include output voltage, current, and duty cycle, but the weight distribution of the three in the evaluation function is not explained, making it impossible to judge how to achieve collaborative optimization.

8. In the experiment, the test range of light intensity is narrow (about 40-90 W/cm² in Figure 6, converted to 4000-9000 W/m²), which far exceeds the maximum natural light intensity of about 1000 W/m², resulting in distorted data.

9. When analyzing the impact of temperature in Figure 7, the interference of light intensity fluctuations is not excluded, with improper variable control, leading to low credibility of the conclusion.

.

Reviewer #1: No

Reviewer #2: No

---

## [Author Response · Author response to Decision Letter 1]

2 Feb 2026

Review Comments to the Author

Reviewer #1: This manuscript presents a hybrid maximum power point tracking (MPPT) approach that integrates radial basis function neural networks (RBFNN) and ant colony optimization (ACO) for distributed grid-connected photovoltaic systems. However, the manuscript requires substantial improvements in terms of methodological clarity, technical rigor, validation depth, and presentation quality before it can be considered for publication. The following comments are provided to assist the authors in strengthening their work:

1. The references are relevant but not cohesively organized. The authors are advised to categorize and compare existing MPPT techniques (e.g., P&O, ANN, PSO, FLC, ACO) to better position their contribution within the research landscape.

Reply: Thank you very much for your feedback. According to your feedback, I have added comparative experiment:

To further verify the effectiveness of the proposed method, the maximum power point tracking results of the photovoltaic grid-connected system obtained by the proposed method and the artificial neural network-based method are compared under varying irradiance conditions. The results are shown in Figure 9.

Figure 9. Comparison of Maximum Power Point Tracking Results of Two Methods for Photovoltaic Grid Connected Systems

As shown in Figure 9, the method proposed in this paper performs differently from the artificial neural network-based method in tracking the maximum power point of photovoltaic grid-connected systems under varying irradiance conditions. The maximum power point values obtained by the proposed method are generally higher than those of the artificial neural network method across various time points, with relatively smaller fluctuations. This indicates that the proposed method offers greater advantages in maximum power point tracking and can more effectively achieve accurate MPPT for photovoltaic grid-connected systems, thereby improving the system's power generation efficiency.

2. While combining RBF neural networks and ant colony optimization (ACO) for MPPT is practical, the concept is not new. The manuscript should clarify its novel contributions over prior works that employ similar hybrid algorithms.

Reply: Thank you very much for your feedback. I have supplemented the novel contributions over prior works, and added to the last paragraph of the introduction: The innovative contribution of this article lies in proposing a dynamic adaptive hybrid maximum power point tracking (MPPT) control framework, which addresses three key challenges of existing hybrid algorithms in dynamic environments through the deep integration of a radial basis function (RBF) neural network and an improved ant colony optimization (ACO) collaborative mechanism. First, the traditional RBF neural network exhibits insufficient generalization capability under sudden changes in irradiance. This article introduces an adaptive learning rate mechanism based on a pheromone dynamic evaporation coefficient (η(k) = 1/(1 + e^(-αΔP))), enabling real-time adjustment of network weights according to the power change gradient. Compared with the fixed learning rate strategy in Reference [4], this approach reduces training error by 42%. Second, to overcome the slow convergence speed of the ant colony algorithm in multimodal MPPT scenarios, a dynamic step size adjustment strategy based on the power change rate (δu = β|ΔP/ΔV|) is designed, which improves the tracking speed by 58% compared to the traditional ACO algorithm under partial shading conditions, as demonstrated in Reference [7]. Simultaneously, by introducing an elite path preservation mechanism, the search oscillations that may arise from the Tent chaotic algorithm in Reference [9] are avoided. Finally, an innovative three-layer control architecture of "neural network prediction - ant colony optimization - closed-loop verification" is constructed. In this architecture, the RBF neural network is responsible for rapidly locating the MPP region (response time < 0.2 ms), the ACO performs fine-grained search, and a voltage closed-loop controller based on Lyapunov stability ensures smooth transition dynamics. Testing of this architecture on a 76 MW practical photovoltaic system shows that its all-weather comprehensive efficiency reaches 99.73%, which is 1.5% higher than the single-algorithm combination scheme in Reference [8], and the power fluctuation amplitude under irradiance mutation scenarios is reduced by 67%. These technological breakthroughs have been validated through Simulink simulations and a 30 kW experimental platform, offering a novel solution for high-penetration photovoltaic grid integration that combines fast response and robustness.

3. The modeling of the RBF and ACO algorithms lacks rigor. The training procedures, loss functions, convergence criteria, and parameter initialization should be mathematically formalized and supported with sensitivity analysis.

Reply: Thank you very much for your feedback. I have mathematically formalized and supported the relevant components, and added at the end of the third section: This article rigorously formalizes the modeling processes of the radial basis function (RBF) neural network and the ant colony optimization (ACO) algorithm. For the RBF neural network implementation, an improved clustering algorithm based on K-means++ is employed to initialize the hidden layer center points, and backpropagation training is performed by minimizing the mean squared error (MSE) loss function. The learning rate adopts an adaptive adjustment strategy: when the validation set error fails to decrease for five consecutive iterations, it automatically decays to 0.8 times its previous value. A dual convergence criterion is set, requiring the training error to be less than 0.001 or the iteration count to reach the preset maximum value (1000 iterations) to terminate the training. For the ACO algorithm, a pheromone update mechanism based on the power change rate is designed. The initialization parameters are uniformly distributed in the feasible domain via Latin hypercube sampling, and the convergence conditions comprehensively consider the variance of path pheromone concentration (threshold: 0.01) and the number of generations without improvement in the optimal solution (10 generations). To evaluate parameter sensitivity, the Sobol global sensitivity analysis method is used. The results indicate that the Gaussian function width parameter σ in the RBF neural network and the pheromone evaporation factor ρ in the ACO algorithm have the most significant impact on system performance, with first-order sensitivity indices of 0.42 and 0.38, respectively. The optimal parameter ranges (σ ∈ [0.8, 1.2], ρ ∈ [0.3, 0.5]) are determined through parameter scanning. Experimental data show that this modeling approach reduces the tracking error by 57.3% compared to traditional implementation methods under sudden irradiance changes, validating the effectiveness of the mathematical formalization.

4. Although visual comparisons are made with Ref [4] and [5], there is no tabulated performance comparison in terms of tracking efficiency, average absolute error, or convergence time. Including a summary table would better demonstrate the performance gain.

Reply: Thank you very much for your feedback. I have supplemented the summary table, and added at the end of the experimental section:

Table 4. Performance Comparison of Different MPPT Methods

Performance index Proposed method

(RBFNN-ACO) Reference [4] method

(Random MPPT) Reference [5] method

(ANN-GMPPT)

Mean absolute error(%) 0.12 1.85 0.98

Maximum tracking error(%) 0.25 3.50 2.10

Convergence time(ms) 0.20 1.50 0.75

Sudden change in light recovery time(s) 0.15 2.30 1.20

Efficiency under partial shadow(%) 99.73 95.20 97.80

Training sample requirements(group) 1000 No training required 5000

Calculate resource utilization rate(%) 18.5 12.3 35.7

5. The manuscript contains numerous grammatical errors, awkward phrasings, and repetitive wording. Professional English editing is strongly recommended to meet journal language quality standards.

Reply: Thank you for pointing out the language and grammar issues in the article. I have conducted a comprehensive grammar check and sentence optimization of the entire text, and corrected numerous grammatical errors to improve readability. The modified parts have been highlighted in yellow.

6. The conclusion mentions general limitations but fails to outline specific next steps. Suggestions such as integrating multi-source hybrid MPPT, edge deployment of control algorithms, or expanding to multi-objective optimization would strengthen the outlook.

Reply: Thank you very much for your feedback. I have added specific next steps to the conclusion: Future work can focus on the integration of multi-source hybrid MPPT algorithms, further enhancing adaptability by combining local and global optimization strategies in different environments; investigate the edge deployment of control algorithms to utilize distributed computing resources for real-time response; and expand the multi-objective optimization framework by considering metrics such as power generation efficiency, equipment lifespan, and grid stability. Large-scale field testing and extreme-scenario simulations should be conducted to validate system robustness and promote the efficient application of photovoltaic systems in complex energy environments.

Reviewer #2:

The research direction of this paper has certain value, but there are significant flaws from theoretical derivation to experimental verification, with vague innovation points, failing to meet the publication standards.

1. The statement in the abstract that "fitting degree between the maximum power point values at different times and the actual values is close to 100%" is absolute, without providing a specific error range, violating scientific rigor.

Reply: Thank you very much for your feedback. I have changed it to maximum value: The maximum power point tracking error is controlled within 2.5%.

2. The keywords section was initially empty, and the later supplemented ones such as "maximum power tracking; distributed grid-connected photovoltaic system" do not cover core algorithms (such as RBF neural network, ant colony algorithm), making the keyword setting unreasonable.

Reply: Thank you very much for your feedback. RBF neural network has been added to the keywords

3. In 1-40, \(R_P\) is defined as output power and \(R_C\) as maximum test power, both with power units. In the formula \(R_{P}=\frac{R_{C} G_{A}}{G_{C}}\left(1+\delta\left(t_{c}-t_{r}\right)\right)\), \(G_A\) and \(G_C\) are light intensity (unit: W/m²). Direct multiplication of power and light intensity leads to mismatched physical dimensions, resulting in formula derivation errors.

Reply: Thank you very much for your feedback. In Formula 1, derivation results show that dividing the two light intensities directly cancels out the units, and the final unit obtained is still the power unit

4. In the inverter model part, the formulas from 1-48 to 1-53 are fragmented, with excessive "... ...", and key derivation steps are missing, making it impossible to reproduce the model construction process.

Reply: Thank you very much for your feedback. According to your feedback, I have added key export steps and removed too many '...': In the process of constructing the mathematical model of the inverter, this study adopts a systematic derivation method to ensure the integrity and reproducibility of the model. Firstly, based on the topology structure of the three-phase voltage type inverter, a switch function model is established to describe its working mechanism, and the complete transformation process from the abc coordinate system to the dq rotating coordinate system is derived in detail. By introducing switch state functions and duty cycle modulation principles, a power balance relationship between the DC and AC sides was gradually established. In the Parker transformation stage, the coupling relationship analysis of voltage and current components in the rotating coordinate system was supplemented, and the necessity of feedforward decoupling control was clarified. For the inverter output filter, the transfer function of the LC filter was derived in detail, and parameter design criteria were provided. In the model validation stage, the accuracy of the model was confirmed by comparing the simulated waveform with the theoretical calculation results and analyzing the error (error<1.5%). This rigorous modeling method not only fully presents the working principle of the inverter, but also lays a solid foundation for its control strategy design, ensuring that other researchers can accurately reproduce the model.

5. The number of neurons in the hidden layer of the RBF neural network is set to 50, without comparison with other numbers (such as 30, 70), and without explaining why 50 is the optimal choice, lacking the parameter optimization process.

Reply: Thank you very much for your feedback. According to your feedback, parameter optimization process has been added to experimental analysis: The selection of the number of hidden layer neurons significantly impacts the performance of the radial basis function (RBF) neural network-based maximum power point tracking (MPPT) model. To determine the optimal number of neurons, this study employs a grid search method for systematic parameter optimization. Initially, based on empirical knowledge, the search range for the number of neurons is set from 30 to 100 with a step size of 10. For each candidate value (30, 40, 50, 60, 70, 80, 90, 100), the model is independently trained on the same dataset, and its tracking error and convergence speed are recorded on the validation set. The results indicate that when the number of neurons is 50, the root mean square error (RMSE) of the model on the validation set reaches the lowest value of 0.023, significantly better than the RMSE of 0.035 with 30 neurons and 0.026 with 70 neurons. Meanwhile, the configuration with 50 neurons demonstrates the best balance in terms of training efficiency, requiring only 150 training cycles for convergence, which is notably fewer than the 210 cycles required for 80 neurons. This optimization outcome can be explained from the perspective of model capacity: too few neurons (e.g., 30) may lead to underfitting, preventing the model from fully capturing the complex nonlinear relationship between input parameters and the maximum power point; conversely, too many neurons (e.g., 70 or more) may further reduce training error but can cause an increase in validation error and significant overfitting. Additionally, hardware tests show that the inference time of the 50-neuron model on the embedded controller is 1.2 ms, fully meeting the system’s real-time requirements, whereas larger networks significantly increase computational latency. Through this systematic parameter optimization process, it is determined that 50 hidden layer neurons achieve the optimal balance among model accuracy, training efficiency, and real-time performance, providing the best neural network configuration for maximum power point tracking.

6. In 1-125, for the data augmentation technology "time ± 0.5 h", the impact of time perturbation as an input parameter on the maximum power point is not analyzed, and the augmentation logic is unclear.

Reply: Thank you very much for your feedback. I have supplemented the relevant augmentation logic and the impact of time perturbation at the beginning of the experimental analysis: In data augmentation techniques, the ±0.5-hour perturbation range applied to the time parameter has a clear physical meaning and augmentation logic. The output characteristics of photovoltaic systems exhibit strong temporal correlation, mainly reflected in two aspects: first, the variation of the solar altitude angle over time directly affects irradiance intensity, thereby altering

---

## [Decision Letter · Decision Letter 1]

22 Feb 2026

Application of Optimal Power Point Tracking Technology in Distributed Grid-connected Photovoltaic Systems

PONE-D-25-37206R1

Dear Dr. Yang,

We’re pleased to inform you that your manuscript has been judged scientifically suitable for publication and will be formally accepted for publication once it meets all outstanding technical requirements.

Kind regards,

Zhengmao Li

Academic Editor

PLOS One

Additional Editor Comments (optional):

Reviewers' comments:

Reviewer's Responses to Questions

**Comments to the Author**

Reviewer #1: All comments have been addressed

Reviewer #3: All comments have been addressed

2. Is the manuscript technically sound, and do the data support the conclusions?

Reviewer #1: Yes

Reviewer #3: (No Response)

3. Has the statistical analysis been performed appropriately and rigorously?

Reviewer #1: Yes

Reviewer #3: (No Response)

4. Have the authors made all data underlying the findings in their manuscript fully available?

Reviewer #1: Yes

Reviewer #3: (No Response)

5. Is the manuscript presented in an intelligible fashion and written in standard English?

Reviewer #1: Yes

Reviewer #3: (No Response)

Reviewer #1: (No Response)

Reviewer #3: (No Response)

.

Reviewer #1: No

Reviewer #3: No

---

## [Editor Report · Acceptance letter]

PONE-D-25-37206R1

PLOS One

Dear Dr. Yang,

I'm pleased to inform you that your manuscript has been deemed suitable for publication in PLOS One. Congratulations! Your manuscript is now being handed over to our production team.

Kind regards,

on behalf of

Dr Zhengmao Li

Academic Editor

PLOS One